# Modelling a Western Lifestyle in Mice: A Novel Approach to Eradicating Aerobic Spore-Forming Bacteria from the Colonic Microbiome and Assessing Long-Term Clinical Outcomes

**DOI:** 10.3390/biomedicines12102274

**Published:** 2024-10-07

**Authors:** Edward Horwell, William Ferreira, Huynh Hong, Philip Bearn, Simon Cutting

**Affiliations:** 1Colorectal Surgical Unit, Ashford & St Peter’s Hospitals NHS Foundation Trust, London KT16 0PZ, UK; e.horwell@nhs.net (E.H.); philip.bearn@nhs.net (P.B.); 2Biomedical Science Unit, Royal Holloway University of London, London TW20 0EY, UK; william.ferreira.2009@live.rhul.ac.uk (W.F.); hong.huynh@rhul.ac.uk (H.H.)

**Keywords:** animal models, spore-forming bacteria, old friends hypothesis, bacillus, gastrointestinal microbiome, metabolism

## Abstract

Introduction: The environmentally acquired aerobic spore-forming (EAS-Fs) bacteria that are ubiquitous in nature (e.g., soil) are transient colonisers of the mammalian gastro-intestinal tract. Without regular exposure, their numbers quickly diminish. These species of bacteria have been suggested to be essential to the normal functioning of metabolic and immunogenic health. The modern Western lifestyle restricts exposure to these EAS-Fs, possibly explaining part of the pathogenesis of many Western diseases. To date, the only animal studies that address specific microbiome modelling are based around germ-free animals. We have designed a new animal model that specifically restricts exposure to environmental sources of bacteria. Methodology: A new protocol, termed *Super Clean*, which involves housing mice in autoclaved individually ventilated cages (IVCs), with autoclaved food/water and strict ascetic handling practice was first experimentally validated. The quantification of EAS-Fs was assessed by heat-treating faecal samples and measuring colony-forming units (CFUs). This was then compared to mice in standard conditions. Mice were housed in their respective groups from birth until 18 months. Stool samples were taken throughout the experiment to assess for abundance in transiently acquired environmental bacteria. Clinical, biochemical, histological, and gene expression markers were analysed for diabetes, hypercholesterolaemia, obesity, inflammatory bowel disease, and non-alcoholic fatty liver disease (the “diseases of the West”). Results: Our results show that stringent adherence to the *Super Clean* protocol produces a significantly decreased abundance of aerobic spore-forming Bacillota after 21 days. This microbiomic shift was correlated with significantly increased levels of obesity and impaired glucose metabolism. There was no evidence of colitis, liver disease or hypercholesterolaemia. Conclusions: This new murine model successfully isolates EAS-Fs and has potential utility for future research, allowing for an investigation into the clinical impact of living in relative hygienic conditions.

## 1. Introduction

Human physiology has co-evolved with the microbial life that make up the colonic microbiome, and their presence is essential to human health [1]. The microbiome includes the bacteria, archaea, fungi, virus, protists, as well as their genomic material and metabolites. They provide a vast repertoire of symbiotic function, with particular importance to metabolic and immunogenic health. Our pre-agrarian ancestors would have phylogenetically evolved in close contact with bacterial species typically found in soil, untreated water, and domesticated animals—for example, members of the phyla Bacillota, Proteobacteria, Actinobacteria and Verrucomicrobia [2,3]. Indeed, spore-forming bacteria (i.e., Bacillota) make up 60% of the colonic microbiota [4]. Moreover, recent studies have suggested that the environmentally acquired aerobic spore formers (EAS-Fs) are highly viable and transmissible to the colonic microbiome, playing an important role in health and maintaining a diverse microbiome [5,6,7,8,9]. These EAS-Fs, often cultured from healthy human faecal samples, have been negatively correlated to the genesis of several human pathologies—such as, inflammatory bowel disease (IBD), asthma, and diabetes [8,10,11,12,13]. The old friends hypothesis is a leading theory that explains the pathogenesis of many chronic diseases seen in the West, linking the absence of these EAS-Fs with the “diseases of affluence. The theory posits that an absence of the bacteria we have co-evolved with leads to an increased risk of metabolic and immunogenic pathology. The modern “Western” lifestyle and diet specifically restricts exposure to these species [14,15,16].

Interestingly, the evidence to date suggests that the aerobic EAS-Fs are *transient* colonisers, and without regular replenishment, the number of viable cells quickly declines [17]. Thus, regular exposure is required to maintain colonies, making the environment a vital reservoir for allochthonous transmission [9,17]. Despite the growing scientific interest in the importance of these environmentally acquired bacteria, very few animal studies have been conducted. 

Due to the ubiquitous nature of spore-forming bacteria, controlling for them on surfaces is difficult and may be an overlooked variable in animal models of disease and are often present in standard animal husbandry conditions [18]. Moreover, due to their heat stability, they can persist despite pasteurisation [19]. Germ-free (GF) animal models have typically been used to model for the effect of specific microbiomes. These conditions allow for the controlled introduction of specific species to observe the clinical impact, providing valuable insights into numerous pathogenic processes [20]. However, GF modelling has the significant limitation of being critically removed from the normal (non-sterile) physiology of humans, and as such, the applicability of the findings becomes limited and hard to interpret. 

Our hypothesis is simple: living in hygienic conditions that restrict the exposure of these transient spore-forming bacterial old friends will increase the risk of developing diseases typified in the West (e.g., obesity, IBD, diabetes, hypercholesterolaemia, and non-alcoholic fatty liver disease (NAFLD)). To address this, we have designed and validated a new animal model that restricts exposure to environmentally acquired bacteria without the need for sterile germ-free conditions. This has been termed *Super Clean*, to distinguish it from the true sterile conditions of GF animal studies. The purpose is to exclude environmentally acquired bacteria and represent the hygienic conditions that are typical of the Western lifestyle, allowing a new technique to model health conditions whilst not hindering the normal physiological development of an animal [21]. We will compare this new model to standard laboratory conditions over an 18-month period and report on several clinical, histological, genomic expression, and biochemical parameters for the aforementioned pathologies. We will comment on how this model can give insights into the role of environmentally acquired bacteria on health and future research potential.

## 2. Materials and Methods

The Animals: C57 BL/6 mice (Charles River, London, UK) were bred in-house, with the mothers being acclimatised to either Super Clean (SC) or standard laboratory (SL) conditions prior to breeding. Female pups were selected and randomly allocated to either SC or SL conditions from the day of birth, with eight pups per group. After reaching weaned maturity (28 days), the mothers were removed, and the litter remained as a group until the end of the study, Figure 1. Due to the length of this study, females were exclusively selected to avoid in-group violence and stress that is typically seen with male-only litter mates [22]. Ethical approval was from our institution’s Animal Welfare and Ethics Review Board (AWERB) and approved by the Home Office (UK). 

*Standard Laboratory Conditions:* As per our standard animal house protocol, mice (*n* = 8) were housed, with up to five mice per cage (MB1, NKP Isotec, Coalville, UK), with ad libitum water (tap) and chow (5LF5, LabDiet, St. Louis, MO, USA). They were exposed to twelve hourly light/dark cycles. Standard bedding, environmental stimulants, and bottles were used (not autoclaved). They were mucked on a weekly basis (*n* = 8). The bedding and environmental stimulants were assessed (prior to use) at three time points throughout the study to quantify EAS-F and ensure no anomalous exposure—the results were consistent throughout the experiment, with a mean heat-treated CFU/g of 2.3 × 10^2^ and 1.8 × 10^2^, respectively.

*Super Clean Design:* The mice (*n* = 8) were housed in groups of up to five mice per unit. They were housed in individually ventilated, HEPA-filtered cages (IVC) (Tecniplast Group, Buguggiate, Italy), with a reported 99.9999937% efficiency at restricting contamination from the environment. To maintain relative sterility, the cage, bedding materials, environmental stimulants, and drinking bottles were all autoclaved prior to use each week. Tap water was autoclaved, and irradiated sterile chow (5L0D, LabDiet, St. Louis, MO, USA) was changed every 2–3 days, both ad libitum. An aseptic protocol was designed for handling the mice during mucking out, employing sterile gloves, with work carried out under a biosafety extraction cabinet. The mothers were housed under SC conditions for 35 days prior to breeding in the attempt to reduce aberrant exposure to environmental bacteria in their offspring. For the quality control of the SC model, intermittent samples of the cage surface, food and water were performed to ensure the absence of environmental contamination, and growth was assessed by plating the heat-treated samples on DSM agar, as described below. This demonstrated no contamination. These IVC cages were housed in a separate room to the SL group but experienced the same light/dark cycles, temperature and humidity. 

*Detection of Aerobic Spore-Forming Bacteria:* Faecal samples were taken, using the clean catch method, at several points throughout the first 100 days of the experiment and at the end of the experiment. These were heat-treated (65 °C, 45 min), serially diluted, and plated on complex media agar (Difco Sporulation Medium (DSM), 37 °C, 24 h). Colony-forming units (CFUs) were calculated and inspected under the microscope to inspect the morphological appearance. 

*Clinical Measurements:* All measurements reported are taken at the end of the experiment when the mice reach 18 months of age, the point at which they are regarded as an old adult [23]. Each mouse was weighed (g), and the naso-anal length (mm) was measured. To classify obesity, the cube root of body weight was divided by the naso-anal length to give the Lee Index (LI) [24], analogous to the body mass index (BMI) used to measure human obesity. Subcutaneous fat was measured (mm) from the thickest part of the posterior inguinal white adipose tissue (WAT), and measurements were an average of the left and right adipose deposits per mouse. To measure visceral fat, the gonadal fat pad was completely excised and weighed (g). A pan-proctocolectomy was then performed and the caecal–anal length was measured (cm). The caecum was then transected from the colon, being careful not to spill its faecal content, and weighed (g). The liver and spleen where harvested and individually weighed (g). 

*Biochemical and Histological Measurements:* A serum sample, from a cardiac puncture, was taken using paediatric K3EDTA tubes and sent for total cholesterol (mmol/L) and alanine transaminase (ALT)(U/L) analysis (Royal Veterinary College, London, UK). A second serum sample was taken for HbA1C (%) measurement (BHR Pharmaceuticals, Nuneaton, UK). Samples of the liver were taken and fixed in 10% (*v*/*v*) neutral buffered formalin. They were histologically processed and stained with H&E and inspected for evidence of non-alcoholic fatty liver disease (NAFLD).

*Gene expression analysis.* The rectosigmoid was sectioned to provide a 100 mg sample, placed in 1 mL TRIzol (Invitrogen, Waltham, MA, USA), and stored at −80 °C for later work. RNA was extracted using chloroform/isopropanol precipitation, as per the manufacturer’s protocol. Residual gDNA was removed and reverse transcription performed (QuantiTect Reverse Transcription, Quigen, Hilden, Germany). Quality was checked using spectrophotometry (Nanodrop, Thermo Fisher Scientific, Waltham, MA, USA). RT-qPCR followed the manufacturer’s protocol (QUantiNova Sybr Green PCR & RotorGene 6000 Thermocycler both from Qiagen, Hilden, Germany). Primer sequences, as seen in Appendix A, were chosen from our previous work on colonic inflammation. To allow a comparative expression fold-change between the SC and SL groups, a third set of samples were taken from 5 female control mice, 14 weeks old, that were housed under SL conditions. 

*Statistical Analysis:* All animals were included in the analysis. GraphPad Prism (Version 9.4.1 for Windows, Dotmatics, Massachusetts, USA) was used. Statistical significance was accepted as *p* < 0.05. Independent-samples *t*-test was employed to determine the statistical difference between the two groups. There were no outliers in the data, as assessed by an inspection of boxplots. All results were normally distributed, as assessed by Shapiro–Wilk’s test. Data are presented as mean ± standard deviation, against the “standard laboratory” group. For RT-qPCR results, the data are transformed logarithmically for statistical analysis and are presented as an expression fold-change to the log2. 

## 3. Results

### 3.1. Super-Clean Conditions Effectively Removed Colonic Spore-Forming Bacteria

The adult breeding mice (prior to breeding) demonstrated a mean of 5.5 × 10^4^ CFU/mL (95% CI = 8.4 × 10^4^–2.7 × 10^4^). For the group that was then placed under SC conditions, this started to decrease after 21 days, reaching a nadir of 2.06 × 10^2^ CFU/mL (95% CI = 293–119) after 35 days of SC exposure, Figure 2A. 

After breeding, once the pups were mature for faecal sampling (~15 days), the SL mice demonstrated a spore count ranging from 4.42 × 10^4^ to 1.06 × 10^5^ CFU/mL. In contrast, the SC group had a significantly lower level of 2.36–4.22 × 10^2^ CFU/mL (*p* ≤ 0.0001; 95% CI 55,320–102,814), with two mice demonstrating zero spores during the experiment, Figure 2B. This quantitative divergence of spore-forming bacteria between the two groups remained significant throughout the experiment, demonstrating that SC conditions successfully removed this group of bacteria from the experiment. When examining the bacteria under the microscope these were predominantly rod-like bacillus, in keeping with the phylum *Bacillota,*
Figure 2C. Furthermore, 16SrRNA analysis was not performed, so we were unable to describe the genus or species composition of this microbiomic shift. 

### 3.2. Weight and Obesity

The SC mice demonstrated a significant increase in total body weight compared to their SL counterparts, with a mean weight of 39.4 g and 31.4 g, respectively (*p* = 0.0003; 95% CI 4.466–11.61), Figure 3A. When calculating the LI, the results remained significant, with the SC group demonstrating a greater body mass index compare to the SL group. For example, the mean SL was 276.3 ∛g/cm and SC was 307.7 ∛g/cm (*p* = 0.0163; 95% CI 6.732–56.07), Figure 3B. 

In keeping with this finding, measurements of subcutaneous fat were markedly increased in the SC group, suggesting the increased weight was due to adipose tissue. The mean gonadal fat pad weight in the SC group was 0.93 g, and mean for the SL group was 0.63 g (*p ≤* 0.0001; 95% CI 0.1843–0.4132), Figure 3C. Similarly, the posterior inguinal adipose thickness was greater in the SC group, with a mean of 1.3 mm, compared to 0.8 mm in the SL group (*p* = 0.0003; 95% CI 0.3223–0.7057), Figure 3D. 

### 3.3. Colonic Measurements

The colonic length of the SL and SC groups were similar, with the mean values being 12.3 cm and 12.6 cm, respectively (*p* = 0.2217), Figure 4A. However, the weight of their respective “wet” caeca was divergent, with a mean value for SL of 2.7 g and SC 3.0 g (*p* = 0.0252; 95% CI 0.03953–0.5105), Figure 4B. 

When analysing the transcriptome of the two groups, there was no difference in gene expression. The two groups’ samples were compared against healthy 14-week-old BALB/c mice, allowing a quantitative analysis. TNF-α was expressed at higher levels in both SL and SC compared to the comparator, with a 3.6 and 3.5 log2 fold increase (*p* = 0.3975), Figure 4C. IL-6 was also found to be upregulated by 3.1 and 3.6, respectively (*p* = 0.6609), Figure 4D. In contrast, IL-1β was downregulated in the SL group by 3.4 and 3.8 in the SC group (*p* = 0.6063), Figure 4E. IL-10 was upregulated in the SL group by 4.3 and 3.1 in the SC group (*p* = 0.2443), Figure 4F. The MUC-2 results for SL and SC were 5.1 and 6.2, respectively (*p* = 0.1297), Figure 4G. The Claudin-1 results for SL and SC were 2.0 and 3.1, respectively (*p* = 0.902), Figure 4H. 

### 3.4. Metabolic Measurements

We found a significant difference in the % of HbA1C between the two groups, with a mean SL value of 4.74% and SC of 5.38% (*p* = 0.0006; 95% CI 0.3270–0.9480), Figure 5A. Furthermore, there was a significant difference in the average hepatic weight, with the SL group being 1.5 g and the SC group 1.8 g (*p* = 0.0151; 95% CI 0.05998–0.4725), Figure 5B. Interestingly, when inspecting the hepatic histology, none of the samples demonstrated non-alcoholic fatty liver disease (NAFLD) or indeed any other pathology, Figure 5C, thus questioning the clinical relevance of this difference. We found no significant difference in serum total cholesterol or ALT levels (*p* = 0.8580 and *p* = 0.2799), Figure 5D,E. We also found no difference in splenic weight (*p* = 0.5152), Figure 5F.

## 4. Discussion

This pilot study has two pertinent findings. Firstly, our new animal model, designed to replicate hygienic conditions of the West by restricting exposure to EAS-Fs, was successful. We found the Super Clean methodology significantly reduced the colony-forming units of these bacteria from the faecal microbiome after 21 days. This partially corroborates former work that has suggested that these species may be transient colonisers [17]. Previous studies have suggested that aerobic spore formers germinate in the small intestine and proliferate for a period of 5–27 days before culturable numbers start to significantly drop unless replenished [25,26,27]. Instead, our data suggest that these species reduce to a “trough level” in the gastrointestinal tract, rather than completely eradicate. This is demonstrated by the persistence seen in the SC group throughout the experiment, with a mean of 3.02 × 10^2^ CFU/mL compared to a mean of 8.178 × 10^4^ CFU/mL in the mice housed under standard conditions. These species were morphologically in keeping with the spore formers of the Bacilli class (e.g., *Bacillus* spp.). This is expected: it has been estimated that over half of the colonic microbiome is comprised of spore-forming bacteria and environmental exposure. 

We believe there are two reasons why the spore counts in the SC group remained at a constant trophic level throughout the experiment: (i) Despite the removal of external sources of these bacterial species, mice are coprophagic animals and may replenish the Bacillota via faecal microbial self-reinoculation. We decided against using a typical grid-like device to prevent coprophagia as these can only limit coprophagia rather than eliminate it. Furthermore, it would add complexity to the SC mucking out protocol, and finally, there is evidence to suggest that mice have several detrimental health outcomes if coprophagia is prohibited [28]. (ii) *Bacillus,* and similar genera, may not be true transient colonisers as previously thought. They may instead occupy a niche in the GIT, allowing a constant attenuated presence that is transiently increased after exposure to external sources. Indeed, several characteristics have been previously identified that would make them good candidates to fill this role, notably the formation of biofilms on the colonic epithelium [29]. Furthermore, another finding from our study supports the observation of reduced bacterial abundance in SC mice—the significantly increased size of the caecum compared to the SL group. It is well established that GF and antibiotic-treated mice have an enlarged caecum that inversely correlates with the level of bacterial abundance due to an accumulation of mucopolysaccharides and water in the caecal lumen [30]. 

The second striking finding is that the reduction in these environmentally acquired bacteria resulted in significant metabolic differences. We found that the SC mice had a mean LI, the murine equivalent to BMI, that was significantly greater than the SL mice. Using the definition for obesity as an LI ≥ 300 ∛g/cm, 75% of SC mice were obese compared to 25% of SL mice. Similarly, subcutaneous fat and visceral fat were found to be significantly higher (both in thickness and weight) in the SC group compared to the SL mice. As with obesity and adipose deposits, we found glycosylated haemoglobin (HbA1C) to be significantly elevated in the SC group, suggesting an impairment in glucose (5.4 vs. 4.7%). HbA1C is a marker of serum glucose (mmol/L) over the prior 60 days in mice (120 in humans) and as such gives a general appreciation for periods of time with relative hyperglycaemia. Furthermore, the SC group mean was also greater than the murine reference range (<4.9%), implying that there could be clinically significant impaired glucose metabolism [31,32]. Interestingly, the obesity and increased visceral fat did not correlate with an increase in total cholesterol or the presence of NAFLD, with both groups having normal results. This raises certain questions as to the translatability of these results to human clinical practice. The absence of steatotosis on the liver biopsy could be resultant of the study being under-powered or it may indicate other metabolic mechanisms that are not captured in our results—perhaps due to the mice eating a *complete diet* rather than a *high-fat diet* typified in the West. Nonetheless, we believe these findings suggest that an absence of exposure to the environmentally acquired transient bacteria results in significant alterations to the microbiome and metabolome of the mice, leading to obesity and hyperglycaemia. This perhaps is not surprising given the increasing evidence base, and interest in, the use of spore-forming bacteria for managing obesity [33,34,35]. We noted that there was an overall relative upregulation in the Th1 inflammatory cytokines, as well as genes related to colonic homeostasis, in both the SC and SL groups. We have attributed this to the aged nature of our experimental mice, a finding that has been published elsewhere [36].

Spore-forming bacteria have the special characteristic of being highly resistant to environmental stress. This is due to their protective soluble proteins, core, cortex, coat, membranes, and soluble proteins [37]. As a result, they can survive prolonged periods of UV radiation, extremes of temperature, the absence of water and nutrition, predation, and the host’s immune system [38]. They are consequently ubiquitous in the natural environment and are highly transmissible. They are less well adapted, however, to the relative anoxic conditions found in the gastro-intestinal tract. As such, regular exposure is required to maintain numbers [25,27]. Others have hypothesised that a Western lifestyle restricts the exposure to EAS-Fs and increases the likelihood of developing chronic metabolic and inflammatory diseases (the old friends hypothesis) [39]. The main genera of EAS-Fs—*Bacillus* and *Clostridia*—are highly metabolically active, producing a plethora of secondary metabolites that have antibiotic, antifungal, antiviral, and immunomodulatory functions and appear to have efficacy in treating several pathologies [7,10,40]. This might explain the epidemiological differences seen in human populations that have regular exposure to EAS-F bacteria and those who do not. To date, we are unaware of any animal models that test this hypothesis and believe this new model will provide invaluable insights, allowing researchers to test and find mechanistic pathways behind the old friends hypothesis. 

There are several limitations of this pilot study that deserve a mention. To culture the aerobic spore-forming bacteria from faeces, heat treatment is required [27]. As such, we are killing all other bacterial and fungal species and might be ignorant to important shifts in the microbiome. Furthermore, a significant limitation to this study is the absence of a metagenomic analysis on the faecal pellets. As such, we are unable to reliably conclude that the clinical effects observed are purely down to these aerobic spore formers, rather than an unobserved species. The reason for neglecting this was twofold. Firstly, spores from the bacteria under examination of this paper, particularly *Bacillus*, are known to be highly resistant to techniques of DNA extraction, with the result of significant under-representation in the metagenomic analysis, resulting in results that are not representative [4]. Secondly, this was a pilot study with the intention of investigating the efficacy in restricting EAS-Fs using the SC protocol and observing metabolic and immunogenic outcomes rather than fully investigating pathophysiological mechanisms. A follow-on study to this work is currently being performed and will examine these effects in more detail, with 16s rRNA metagenomic analysis, which may provide more insights into these microbiome-mediated clinical effects. Furthermore, in our approach to replicate the hygienic conditions of the West by using the SC protocol, we created an extreme example of relative sterility that may not fully translate to real-world conditions. However, we believe that the model is substantially more generalisable than it would be using germ-free mice. Moreover, there are significant improvements in economic and practical viability for researchers. 

Despite these limitations, if the results from this study are taken at face value, that exposure to EAS-Fs is diminished in a modern Western lifestyle with implications for long-term health, the logical next question is how to does one increase their exposure to these beneficial microbes? Surprisingly, very little research has been carried out on this specific topic. It has been suggested that organic agricultural techniques produce arable crops with greater microbial diversity, which may provide these beneficial microbes [41,42]. Furthermore, there is growing recognition that exposure to “green spaces” provides EAS-Fs, amongst other beneficial species, that modulate the colonic microbiome [43,44,45]. This appears largely due to soil-based bacteria. As such, a recent study from Finland has shown that transplanting soil and vegetation from a forest into an inner-city nursery significantly altered children’s microbiome, with advantageous immunoregulatory effects [46]. Clearly, more research is required on how we interact with these environmental species and what interventions can be carried out to increase our exposure whilst living a Western lifestyle.

## 5. Conclusions

After several weeks of strict adherence to the “*Super Clean*” experimental conditions, environmentally acquired spore-forming bacteria are significantly reduced from the murine colonic microbiome, becoming evanescent. These experimental conditions, designed to reflect the Western lifestyle, allow a more accurate examination on the long-term health outcome from this environmental variable than has been possible with previous animal models. We found that the lack of these bacteria directly correlates to poor long-term health, resulting in an analogous metabolic syndrome that is commonly seen in the West.

## Figures and Tables

**Figure 1 biomedicines-12-02274-f001:**
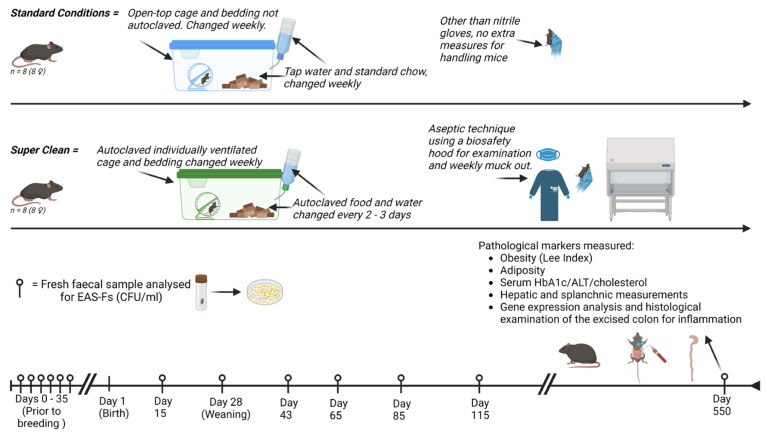
Schematic of experimental design and timeline.

**Figure 2 biomedicines-12-02274-f002:**
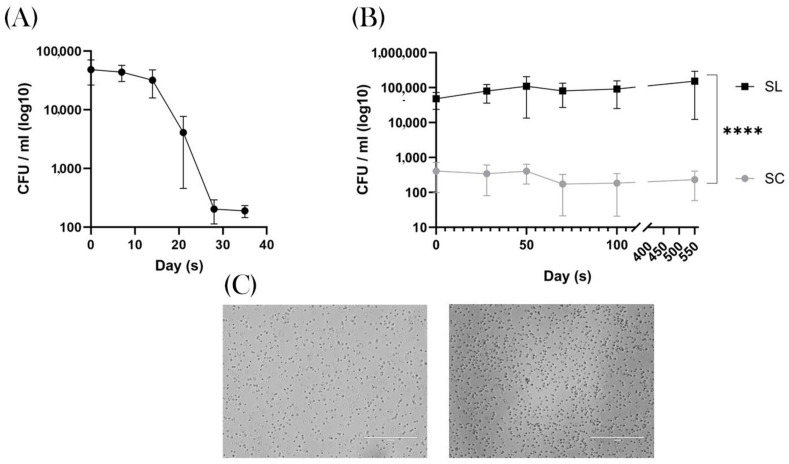
Colony-forming units. (**A**) Faecal samples taken during a five-week acclimatisation period. The graph shows SC conditions produce a decline in heat-resistant aerobic spore formers after three weeks, reaching a plateau of 2 × 10^2^ CFU/mL. (**B**) Samples taken at regular intervals over a 100-day period in the weaned mice that were born into SC (grey) and SL (black) conditions. The results show a significant difference in CFU/mL that is constant throughout the experiment. **** *p* ≤ 0.0001. (**C**) Representative images of the typical morphology of the observed spore-forming bacteria, in keeping with Bacillota.

**Figure 3 biomedicines-12-02274-f003:**
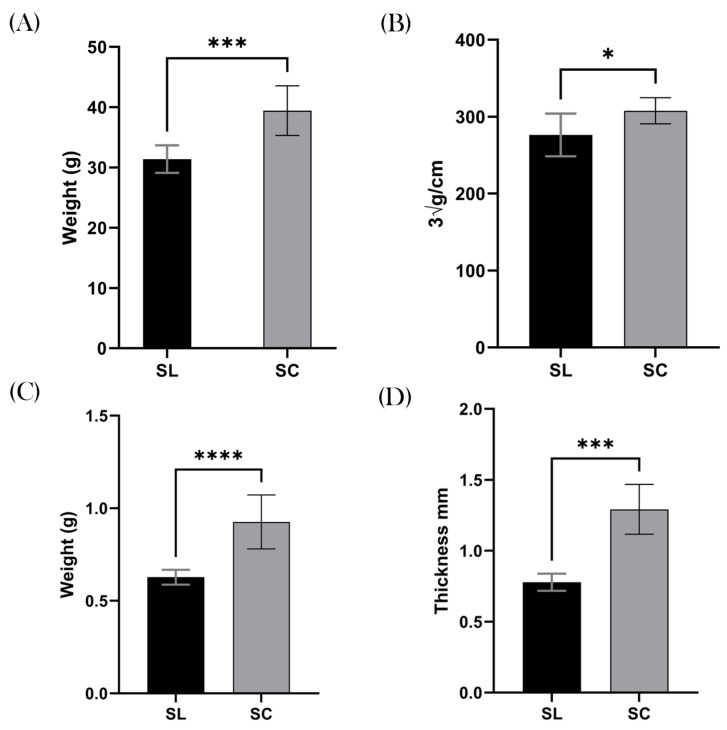
Obesity. All clinical measurements for obesity and adipose tissue demonstrated increased quantities in the SC group compared to the SL group. (**A**) The mean total body weight (g) of the SL group was significantly lower than the SC group. (**B**) This graph shows that the SC group had significantly higher levels of obesity, as measured by the Lee Index (∛g/cm). (**C**) The mean weight of the gonadal fat pad (g) was higher in the SC group. (**D**) The mean thickness of the inguinal adipose tissue was also markedly increased in the SC group compared to the SL group. ns = non-significant, * *p* ≤ 0.05, *** *p* ≤ 0.001, **** *p* ≤ 0.0001.

**Figure 4 biomedicines-12-02274-f004:**
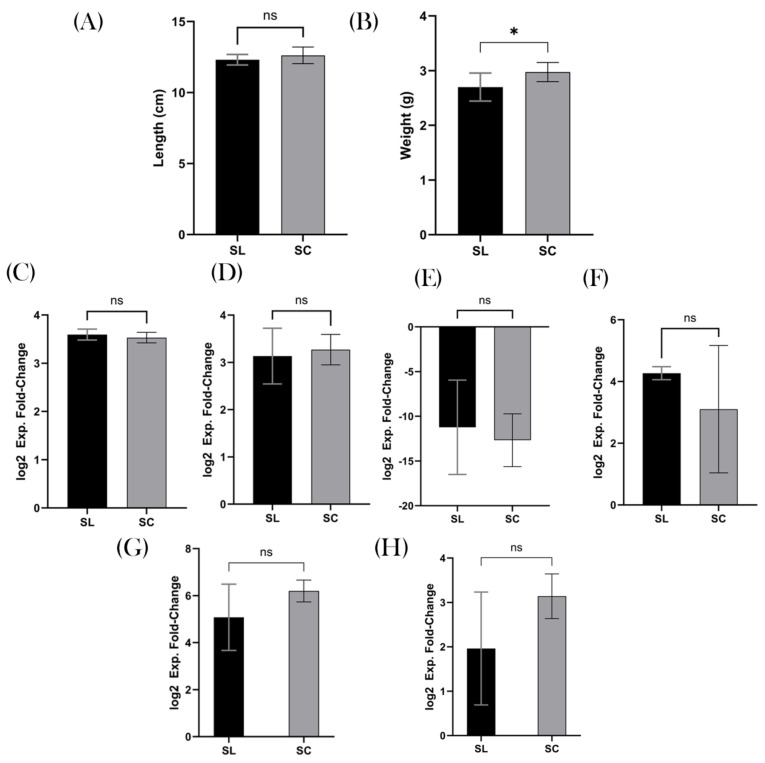
Colonic inflammation. (**A**) Colonic length (cm), a surrogate marker for inflammation in mice, showed no significant difference between the groups. (**B**) Wet caecal weight (g) was, however, increased in the SC group, suggesting decreased microbial diversity. RTq-PCR results demonstrated no difference between the groups, with (**C**) representing TNF-α; (**D**) IL-6; (**E**) IL-1β; (**F**) IL-10; (**G**) MUC-2l and (**H**) Claudin-1. ns = non-significant, * *p* ≤ 0.05.

**Figure 5 biomedicines-12-02274-f005:**
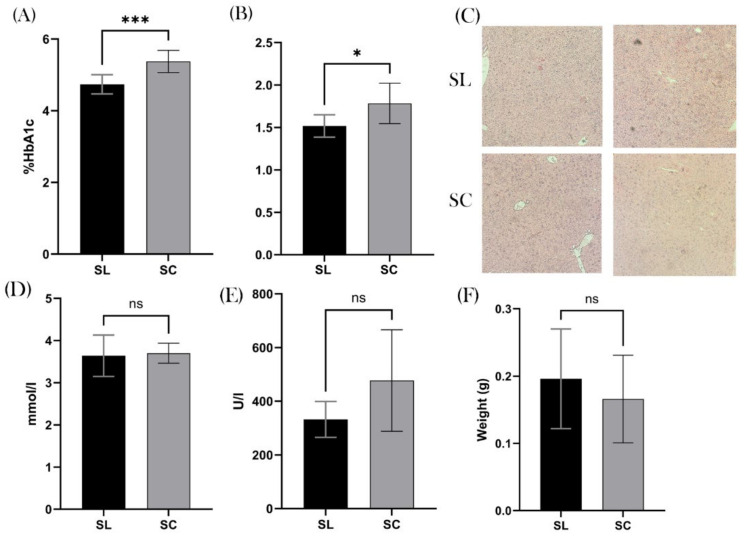
Metabolic health. (**A**) Significantly increased HbA1C in the SC group (5.38%) compared to the SL group (4.74). (**B**), The mean weight of the liver was also significantly increased in the SC group. (**C**) Histological slides of the liver with H&E staining demonstrates, however, that this increased weight was not due steatosis. (**D**) There was no difference in serum total cholesterol (mmol/L). (**E**) Similarly, no significant difference in serum alanine transaminase (ALT) (U/L) was shown. (**F**) The mean weight of the spleen (g) was also the same between the two groups. ns = non-significant, * *p* ≤ 0.05, *** *p* ≤ 0.001.

## Data Availability

All data are in the manuscript and figures. The raw data generated from this study are available upon request to the corresponding author.

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
