# Peer review of "Modelling a Western Lifestyle in Mice: A Novel Approach to Eradicating Aerobic Spore-Forming Bacteria from the Colonic Microbiome and Assessing Long-Term Clinical Outcomes"

_biomedicines, 2024, doi:10.3390/biomedicines12102274_

Round 1

Reviewer 1 Report

Comments and Suggestions for Authors

Dear authors 

Thank you for the excellent manuscript. The idea is good and presented well.

Introduction

Is very good and give a broad scope about the importance of environmentally acquired aerobic spore-forming to human health and the impact of Western lifestyle on EAS-Fs.   

Materials & Methods 

The Super Clean protocol used in the research is a new model and the authors success in using it instead of Germ-free (GF) animal models. 

This part is writen well. But I think the 16rS and metagenomic could give more evidence to the study results and it's one of the most important limitation of the study.

Results and discussion are very good. Figures are good.

Figure 5 (F) The symbol does not appear in the figure 

The results of using SC as model mimic to Western lifestyl was succes in inducing obesity and hyperglycemia in mice.  

Author Response

Thank you for your review. 

We have made a revision to acknowledge the shortcoming of no metagenomic analysis, and how to address this in future work

Thank you for spotting the error with figure 5 - this is now fixed. 

Reviewer 2 Report

Comments and Suggestions for Authors

See pdf document.

Author Response

Thank you for your review. 

Introduction - this is an interesting thought. There is very limited research specifically to EAS-Fs, but I have included a comment on what data is available. 

Materials and methods - 

3) This is a good point. We measured the bedding and environmental stimulants at three different time points. The results were consistant, and I have included this into the methodology. 

4) The two different cages are housed in different rooms, but under the same conditions (i.e. temperature, light/dark cycle times, and humidity). However, the Technoplast IVC cages are essentially hermetically sealed from the outside environment (other than light). The humidity and temperature of the air was set to be identical to the standard cages. 

I will include a description of this in the methodology to make the point clearer for reproducibility. 

5) Both diets are based on LabDiet's "Constant Nutrition" formula. The guaranteed analysis for both of the diets are the same. The margin of error for this guaranteed analysis for 5L0D and 5LF5 is 1% off, which we believe makes them equivalent. 

6) This is as per the Trizol (Invitrogen)'s protocol that was referenced in the previous sentence. I could reference the original 1987 Chomczynski and Sacchi paper if this is what you mean? 

7) I agree it is missing information that will be very interesting to know. This study however was purely designed as a pilot to test the methodological efficacy of excluding EAS-Fs and basic clinical markers for inflammatory and metabolic change. It was not set-up to investigate the underlying detailed microbiome and physiological changes that occur as a result. The results from this pilot have however meant that a more detailed follow on study is now currently underway - and here we are going to be spending particular attention to full microbiomic and metabolomic analysis. Another reviewer has made the same point, and this limitation of the work has been expanded upon in the discussion. 

8) Thankyou for this recommendation. It is not a methodology I am familiar with, but we I will read up on it and take it into serious consideration for the follow on study. 

Reviewer 3 Report

Comments and Suggestions for Authors

The manuscript presents a novel and innovative approach to studying the impact of a Western lifestyle on the gut microbiome, with a focus on environmentally acquired aerobic spore-forming (EAS-F) bacteria. The research provides a unique perspective by examining the potential link between reduced exposure to these bacteria and the development of diseases prevalent in Western societies, such as obesity, diabetes, and inflammatory bowel disease. The introduction of a new murine model, which mimics the limited bacterial exposure typical of Western lifestyles, offers significant promise for advancing understanding in this field.

The study's relevance is clear, and the findings may contribute substantially to the growing body of research on the gut microbiome and its role in human health, particularly in relation to Western lifestyle diseases. The focus on EAS-F bacteria, a relatively underexplored area, adds further originality to the study, making it an important contribution to both microbiome research and public health discussions.

1. The introduction provides a good overview of the study’s goals, but it could benefit from a more detailed background on the microbiome. Incorporating some introductory facts about the microbiome would help readers unfamiliar with the field understand the significance of the gut bacteria being studied.

2. My primary concern is that, while the study demonstrates significant increases in body weight, fat mass, and impaired glucose metabolism in SC mice, the absence of MAFLD and normal cholesterol levels calls into question the clinical relevance of these findings. The discrepancy between the observed metabolic changes and the lack of associated conditions such as MAFLD and dyslipidemia suggests that the implications of the metabolic alterations might not fully translate to clinical scenarios or might be indicative of other underlying factors not addressed in this study.

3. There are some minor technical issues with the formatting of references. The manuscript should ensure that the citation style aligns with the journal's guidelines.

In conclusion, the manuscript is well-conceived, timely, and has the potential to significantly impact research into the gut microbiome and its relationship with Western diseases. Therefore, I recommend minor revisions before publication.

Author Response

Thank you for your time in reviewing our paper and providing positive and constructive comments. 

1) Within the limitations of an introduction, I am concerned that going into more detail on the microbiome than is already written could create a mini-literature review within the introduction itself - running the risk of distracting the reader as to the purpose of the study.

We of course can insert more details if you really think this is necessary. Was there any aspect in particular that you felt lacking? Perhaps, a sentence guiding readers to the very well written review article (https://doi.org/10.1038/nm.4517) would be sufficient? 

2) This is a good observation. There are a few considerations to keep in mind though. The C57 mice were not fed a high-fat diet that would typically induce NALFD. They were given a complete life cycle (balanced) diet that does not obesogenic, and as such we are not hugely surprised that this has not translated to significant steatosis in the biopsies.

Nonetheless, your concern is valid. This study was not designed to show any underlying physiological mechanism. Rather it was designed to assess the efficacy of excluding EAS-Fs and record immunogenic and metabolic differences.

The manuscript has been edited to make this point more clear in the discussion so there is no confusion. For future note, a follow on study is currently underway (18 month study time) that will address these mechanistic effects in detail. 

3. Thankyou for pointing this out, it is now addressed.